# Physical Activity and Glycemic Control Status in Chinese Patients with Type 2 Diabetes: A Secondary Analysis of a Randomized Controlled Trial

**DOI:** 10.3390/ijerph18084292

**Published:** 2021-04-18

**Authors:** Wei-Yuan Yao, Meng-Ge Han, Giuseppe De Vito, Hong Fang, Qinghua Xia, Yingyao Chen, Xiaona Liu, Yan Wei, Russell L. Rothman, Wang-Hong Xu

**Affiliations:** 1Key Laboratory of Public Health Safety (Ministry of Health), School of Public Health, Fudan University, Shanghai 200032, China; 19211020004@fudan.edu.cn (W.-Y.Y.); 16211020051@fudan.edu.cn (M.-G.H.); 14211020005@fudan.edu.cn (X.L.); 14111020034@fudan.edu.cn (Y.W.); 2Department of Biomedical Sciences, University of Padova, 35131 Padova, Italy; giuseppe.devito@unipd.it; 3Minhang District Center for Disease Control and Prevention, Shanghai 201102, China; mhcdcfh@126.com; 4Changning District Center for Disease Control and Prevention, Shanghai 200051, China; xiaqinghua56@126.com; 5Key Laboratory of Health Technology Assessment (National Health Commission), School of Public Health, Fudan University, Shanghai 200032, China; yychen@shmu.edu.cn; 6Department of Medicine, Vanderbilt University Medical Center, Nashville, TN 37232, USA; russell.rothman@vanderbilt.edu

**Keywords:** randomized control trial, type 2 diabetes, physical activity, glycemic control

## Abstract

This secondary analysis was designed to evaluate the independent effect of physical activity (PA) on hemoglobin A1c (HbA1c) level in Chinese patients with type 2 diabetes mellitus (T2DM). A total of 799 T2DM patients from eight communities of Shanghai, China, were randomized into one control arm and three intervention arms receiving 1-year interventions of health literacy, exercise, or both. PA was measured using the International Physical Activity Questionnaire at baseline, 12 months, and 24 months and quantified as metabolic equivalents (Mets). A multiple level mixed regression model was applied to evaluate the associations between PA and HbA1c level. After adjusting for potential confounders including interaction of PA level with initial PA or HbA1c, a significant improved HbA1c was observed for the patients in the medium versus the lowest tertile groups of overall PA at 12 months (β: −3.47, 95%CI: −5.33, −1.60) and for those in the highest versus the lowest tertile group at 24 months (β: −0.50, 95%CI: −1.00, −0.01), resulting in a β (95%CI) of −3.49 (95%CI: −5.87, −1.11) during the whole two-year period of follow-up. The negative association was also observed when the subjects were classified according to their exercise levels using the World Health Organization (WHO) recommendation as a cut-off point. The beneficial effect of higher PA level was only observed among patients having a lower level of baseline HbA1c or PA or both (all *p* values for interaction <0.05). Our results provide evidence for the beneficial effect of PA and suggest that the exercise intervention should be addressed to the physically inactive patients to improve their PA level to a physiological threshold.

## 1. Introduction

Type 2 diabetes mellitus (T2DM) is one of the most common non-communicable diseases around the world [1]. Self-management behaviors such as diet control, physical activity (PA), medical adherence, and self-glucose monitoring have been suggested to achieve better glycemic control status and clinic outcomes and thus reduce substantial physical, psychological, and socioeconomic burdens of the disease at both family and society levels [2,3].

PA was defined as any body movement produced by skeletal muscles that results in energy expenditure [4]. The healthy and functional benefits of being active for T2DM patients have been well recognized, although the value of diet and exercise was not that helpful in the LOOK Ahead Trial [5]. Regular PA helps to maintain healthy weight and metabolic balance, improves cardiorespiratory fitness, lipid profiles, and musculoskeletal health, and thus reduces incidence of complications and all-cause mortality in T2DM patients [6,7]. In addition to leisure time physical activities, occupational, commuting, and domestic activities were also equally beneficial for human health [8,9].

Possibly due to its delayed health effect, the need for persistent commitment, sweating, unpleasant feelings, and even pain [10,11], regular exercise is one of the most difficult self-management behaviors to follow for T2DM patients. Adults with diabetes are recommended to engage in at least 150 min of moderate to vigorous-intensity exercise per week [12,13]. However, epidemiological studies suggest that most patients were insufficiently active, with only 23–37% patients reaching the recommended exercise level in the USA, 21% in Canada, and 15.3% in mainland China [14,15,16]. Numerous strategies have been proposed to increase the adoption and maintenance of regular PA in T2DM patients. However, there is still a big gap in identification of the best PA intervention that can maximize glucose control and be sustained over the long term [17]. In recent years, a novel intervention was designed focusing on the health literacy (HL) of diabetes patients, namely, on patients’ abilities to obtain, understand, and communicate basic information needed to make appropriate health-related decisions [18]. The HL-oriented intervention has been observed to improve glycemic control status effectively by enhancing self-management behaviors [19,20].

In a cluster randomized controlled trial (RCT) conducted in Shanghai, China, we found that both HL and exercise-focused interventions could improve hemoglobin A1c (HbA1c) level in Chinese patients with T2DM [21]. In this study, the PA level of the patients, not only for leisure-time activities but also for commuting and domestic activities, could be improved by the exercise-focused intervention and the exercise module of the PRIDE toolkit used in the HL intervention (Appendix A). Therefore, the independent effect of PA level on glycemic control remained unclear.

To evaluate the improved PA levels by the interventions, and the effect of PA on improvements in HbA1c, we conducted a secondary analysis of the data derived from the trial conducted in Chinese T2DM patients.

## 2. Materials and Methods

### 2.1. Study Design and Subjects

This study was a secondary analysis of a cluster RCT registered with the International Standard Randomized Controlled Trial (Trial registration: ISRCTN76130594; registration date, 12 January 2015). The trial was designed to evaluate the effectiveness of a comprehensive HL strategy, an exercise-focused intervention, or a combination of both on glycemic control and other outcomes in Chinese T2DM patients. A full description of the trial, including the methods for participant recruitment, has been published previously [22]. Briefly, eight Community Healthcare Centers (CHCs) were selected by convenience from a total of 26 CHCs, with 4 from Minhang District and 4 from Changning District of Shanghai, China. From each CHC, 3 to 5 clinic sites were selected. All clinical sites meeting the following criteria were selected from each center: (i) at least 20 patients can be recruited per site; (ii) a general practitioner (GP) team including at least 2 to 4 physician(s), nurse practitioner(s), or diabetes educator(s) per site can participate in the intervention; (iii) agree to participate for a minimum of 2 years; and (iv) agree to be randomized to any arm of the study.

A total of 799 diabetes patients were recruited from the 35 selected clinic sites during the period of February 2015 and March 2016, as described in our previous report [21]. The inclusion criteria for participants included clinical diagnosis of T2DM, 18 to 85 years of age, most recent HbA1c ≥ 7.5% and/or fasting glucose level ≥10 mmol/L, and willing to participate in the project for the full 2-year duration. Patients were excluded if they had poor visual acuity (vision worse than 0.1/4.0 using the Standard Logarithmic Visual Acuity Chart), significant dementia or psychosis (by health provider report or chart review), or terminal illness with anticipated life expectancy less than 2 years.

Randomization occurred at the level of the Community Healthcare Centers. Six Centers (35 clinic sites) were randomized to receive interventions, with two centers (9 clinic sites) focusing on HL-oriented intervention, two (9 clinic sites) focusing on exercise intervention, and two (9 clinic sites) focusing on comprehensive interventions, while two centers (8 clinic sites) were randomized into the control condition.

Ethics approval was obtained from the Medical Ethics Committee of Fudan University (IRB00002408 & FWA00002399) before recruiting study participants (registration number: 2013-06-0451). All local medical ethics committees agreed with this approval according to the ethical standards of the Declaration of Helsinki, and all participants provided written informed consent. This study followed the CONSORT guidelines for reporting [23].

### 2.2. Interventions

The control group received usual diabetes care according to the Chinese National Guidelines [24], which includes conventional clinical consultations and treatment provision according to existing knowledge and at the individual clinician’s discretion.

Patients in the exercise group received usual diabetes care and were asked to walk 3–5 times a week, 30–40 min per time in the first 6 months and 60–70 min per time in the following 6 months. They could walk continuously or divide the exercise into shorter time periods but for at least 10 min at a time. The intensity of exercise was ideally kept between 12 and 15 in the Borg RPE (Rating of Perceived Exertion) visual scale [25]. In addition, the patients in the group were required and trained by the GPs to record the frequency, time, and intensity (Borg scores) of walking in calendar books specially designed for the trial during the one-year period of interventions for the purpose of supervision.

In addition to usual diabetes care, the intervention for the HL group included use of a Chinese adapted version of the Partnership to Improve Diabetes Education (PRIDE) toolkit to aid healthcare provider–patient communications about diabetes management and a Clear Health Communication Curriculum for healthcare providers to improve diabetes-related counseling communication skills, with specific attention to issues of literacy and numeracy. The PRIDE toolkit has 24 educational modules covering all components of diabetes self-management including diet, exercise, foot care, glucose monitoring, medication management, etc., which was written at a low literacy level with many pictures, white space, and other accommodations for low literacy patients [26,27]. The exercise module includes information on benefits of being active, fun ways to be more active, the dos and don’ts in exercise, as well as goal-setting tasks (Appendix A). The Chinese version of the materials was used by health care providers in the HL group during regular patient-related visits at least 3 times a month, 5 to 7 min per time. At each visit, the providers were asked to cover at least two components from the toolkit materials as well as to perform and document at least one goal-setting task with the patient.

Before initiation of the HL intervention, providers gathered to obtain an approximately 5 to 6-h training on diabetes management, introduction of the Chinese version of the PRIDE toolkit, clear health communication skills, and application of the toolkit using principles of clear health communication. A post-training certification process was performed to ensure that providers had known how to use the materials.

Patients in the comprehensive intervention group received usual diabetes care and both HL and exercise interventions.

### 2.3. Data Collection and Assessment of PA Level

Information on demographic characteristics, diagnosis of diabetes, lifestyle factors including dietary habits (measured by 3-day 24-h dietary recall), and anti-diabetes agent use was collected at baseline through in-person interviews. All subjects were followed up at the end of 1-year interventions and 1-year post-interventions (Figure 1). HbA1c level was measured at baseline and at each follow-up survey using point-of-care high-performance liquid chromatography available in each Community Healthcare Center.

Considering that both HL and exercise-focused interventions were designed to improve PA level by integrating the activities into daily life, we assessed the PA level using the validated International Physical Activity Questionnaire (IPAQ) [28] instead of a walking specific IPAQ [29] at baseline as well as 12- and 24-month follow-up. Leisure-time activities included moderate- and vigorous-intensity exercises, which were defined as activities that cause a small increase in breathing or heart rate such as dancing, yoga, or tai chi, and exercises causing large increases in breathing or heart rate such as running, playing basketball, boxing, etc., respectively. Commuting activities included bicycling and walking during commuting, exercise, or shopping that last at least 10 min. Domestic activities included several key household tasks such as cooking or preparing food, washing dishes, doing laundry, cleaning the house, and child care that last at least 10 min each time. Sedentary time included time spent on TV watching, computer using, video game playing, and reading at home, in the car, or with friends, but not during working. Participants were asked about the frequency and average time spent for each type of PA in a typical week.

The PA level was quantified as metabolic equivalent (MET)-hours/week by multiplying the METs, duration, and frequency of activities from exercise, transportation and housework [30,31]. The value of METs for each PA was recommended by the World Health Organization (WHO) guideline: 4 METs for moderate exercise, 8 METs for vigorous exercise, 4 METs for both cycling and walking, and an average value of 3 METs was assigned to calculate the energy expenditure in domestic activities according to the Compendium of Physical Activities [30,32].

### 2.4. Reclassification of Subjects by Changes in PA Level

All subjects were classified into three groups by the tertile of overall PA levels at 12 months and at 24 months, respectively. We also reclassified the participants into two groups using the WHO recommended level for exercise (10 METs-hours/week or 600 METs-mins/week) as the cut-off point [32].

### 2.5. Statistical Methods

Continuous variables were presented as mean and standard deviation (SD) or medians and interquartile range (IQR), while categorical variables were presented as frequency and percentages. Comparisons of baseline characteristics by intervention status and PA groups were conducted using one-way analysis of variance (ANOVA) or Kruskal–Wallis tests for continuous variables and chi-square tests for categorical variables. Multiple-level mixed regression models were applied to estimate the associations of PA level with HbA1c after adjusting for age, sex, duration of diabetes, anti-diabetes drug or insulin use, monthly income level, initial levels of HbA1c and PA, and two-way or three-way interactions of PA with initial levels of PA and HbA1c. Since no significant difference was observed in characteristics between participants and those lost to follow-up, all missing values (95 at 12 months and 147 at 24 months) were excluded from the analyses. Moreover, we did not observe a significant heterogeneity in associations between PA and HbA1c by sex; we conducted analyses in all subjects.

All analyses were conducted using SAS 9.4 (SAS institute, Inc., Cary, NC, USA). Two-sided *p*-values < 0.05 were considered statistically significant.

## 3. Results

Of a total of 799 study participants, complete data were available for 704 patients (88.1%) at the end of intervention (12 months) and 652 patients (81.6%) at 1-year post-intervention (24 months). Appendix A shows the baseline characteristics of all participants by intervention status. No significant difference was observed across the four groups on age, sex, tobacco and alcohol use, duration of diabetes, and HbA1c level. However, the four groups were not comparable on education, HL and numeracy levels, monthly income per capita, and use of anti-diabetes agents.

Regarding the PA level, the overall and specific type of PA levels were significantly higher in the control arm than in the three intervention groups, as shown in Table 1. Further analysis showed higher baseline levels of PA in participants with lower HbA1c levels regardless of PA type, but the differences reached significant only for commuting activities in women (Appendix A).

The significant effects of HL and exercise interventions on HbA1c level have been described in our previous report [21]. After adjusting for the intervention status, the improvements in HbA1c during the two-year period of follow-up were significantly different by monthly income per capita and years of diabetes. No significant difference was observed in improvements by age, sex, educational level, tobacco and alcohol use, medications, HL level, and PA level (Table 2).

As shown in Table 1, the overall PA level in control arm remained unchanged during the whole 2-year observation. In the exercise group, PA level was observed to increase from baseline to post-intervention (*p* < 0.001) but decreased significantly to the baseline level during the next 12 months (*p* > 0.05). In the HL group, the overall PA level decreased significantly from baseline to post-intervention (*p* < 0.001) and remained unchanged thereafter. The PA level did not change significantly during the 1-year intervention in the comprehensive group, but it increased significantly during the 1-year post-intervention.

We reclassified all participants into three groups according to tertile of PA levels at 12 months (at the end of 1-year intervention) and at 24 months (1-year post-intervention), respectively. As presented in Table 3, the three groups classified by PA level at two time points differed in age, sex, alcohol drinking, hypoglycemic medications, and PA level at baseline.

Further analysis show a significant improvement in HbA1c for the medium versus the lowest tertile group of PA level at 12 months (β: −3.47, 95%CI: −5.33, −1.60) and for the highest versus the lowest tertile group of PA at 24 months (β: −0.50, 95%CI: −1.00, −0.01) after adjusting for potential confounders including significant interaction of PA with initial levels of HbA1c and PA (Table 4). As a result, during the whole 2-year period, a significant improvement in HbA1c was observed for the highest versus the lowest tertile group of PA (β: −3.49; 95%CI: −5.87, −1.11), and a significant interaction was found for PA with baseline levels of PA and HbA1c (*p* for interaction < 0.05).

Figure 2 shows the potential effect of specific type of PA on HbA1c level. After adjusting for overall PA and other potential confounders, a significant increase in HbA1c was observed for the highest versus the lowest tertile group of housework from baseline to 12 months (β: 0.47, 95%CI: 0.07, 0.86). No significant association was observed for other type of PA with HbA1c level.

Stratified analyses by the medians of baseline HbA1c and PA levels were conducted to demonstrate the potential joint effect of PA with the two factors. As shown in Figure 3, the associations of PA with HbA1c level differed by baseline levels of PA and/or HbA1c (all *p*-values for interaction <0.05). An improved HbA1c level was associated with a higher level of PA only among patients with lower levels of baseline PA (<59.5 MET-hours/week) and HbA1c (≥8.1%), with β (95%CI) for the medium and the highest tertile groups versus the lowest group being −0.21 (−0.64, 0.22) and −0.05 (−0.54, 0.44) from baseline to 12 months, and being −0.23 (−0.57, 0.12) and −0.30 (−0.68, 0.09) from baseline to 24 months. In other groups with a higher baseline PA or HbA1c level, a positive association was observed between PA and HbA1c level.

## 4. Discussion

In this secondary analysis of data from an RCT including community-dwelling Chinese diabetes patients, we compared the effectiveness of the exercise, HL, and comprehensive interventions on PA levels from three major domains, and we further evaluated the potential effect of PA on HbA1c level in the population. Our results suggest that an exercise-focused intervention may increase the PA level in Chinese diabetes patients, and the effect of PA on HbA1c depends on the baseline level of PA and/or HbA1c.

At the beginning of the interventions, 39.9% of our patients met the 2018 exercise guidelines for people with diabetes, which recommends at least 150 min of moderate- to vigorous intensity aerobic exercise each week (10 MET-hours/week). As a result, the mean and median levels of baseline PA in our subjects were 67.9 and 59.5 MET-hours/week (4075 and 3570 MET-mins/week), respectively, which is much lower than 213 MET-hours/week (12,780 MET-mins/week), the average national level in Chinese adults in 2009. The sedentary lifestyles in our subjects may account, at least in part, for their poor glycemic control status.

Most previous studies observed the role of exercise intervention in increasing PA levels in T2DM patients [33,34,35], but a null result was usually observed for a health education intervention [36]. For instance, Araizi et al. [35] reported a significantly increased PA level in the active group who were instructed to walk at least 10,000 steps per day 5 or more days per week comparing with the control group in a 6-week RCT. Olson et al. [33] found that the aerobic exercise (walking) intervention increased PA level at post-intervention (month-two), but this was followed by a decline at six months. They did not observe any change in PA level across the period in an education group who were provided with an 8-week online diabetes and health education course. Consistent with these studies, we did not observe changes in PA level in the HL intervention group but found that an exercise-focused intervention—supervised walking—increased PA level in our subjects. However, the effect was not sustained during the post-intervention period, suggesting that continuous exercise interventions may be needed to maintain PA level in the population. Our results, as well as Olson et al.’s report [33], suggest the limited effect of health education on individuals’ PA behaviors and the importance of supervised behavior interventions.

Interestingly, although HL intervention alone did not improve PA level, the comprehensive group receiving both HL and PA interventions had a higher PA level during the post-intervention period, suggesting the long-term effect of combined use of exercise and HL interventions. It seems that HL intervention may help to promote autonomous motives [37] and increase adherence to exercise intervention in our subjects. The adherence was influenced not only by the intervention package used but also by the factors such as social support, physical fitness, socioeconomic status, knowledge of diabetes, infrastructure provision, and integrated policies [38,39,40]. Therefore, continuous exercise intervention or combined use of exercise and HL interventions should be provided to diabetes patients to establish healthy behaviors for long-term benefits.

Diabetes patients are usually encouraged to be active, which has been beneficial for glycemic control. In a meta-analysis, Avery et al. [41] demonstrated a significant negative relationship between increase in PA intensity and HbA1c level, highlighting the importance of physically active behaviors. However, evidence was far from consistent. Mynarski et al. [42] did not find a significant correlation between HbA1c level and PA level in T2DM patients. Based on the observed limited impact of PA on glycemic/metabolic control, Wisse et al. [43] made a conclusion that a PA program was not sufficient to improve glycemic control in T2DM patients. It seems that the functional thresholds, in terms of both the volume and intensity of the delivered PA intervention, may exist and are needed for glucose control. Van Dyck et al. [44] found that HbA1c was improved significantly only among diabetes patients who increased more than 4000 steps per day. In a cross-sectional study, Li et al. [45] found that HbA1c was significantly lower in the sufficiently active group (10–50 Met-hours/week) and very active group (>50 Met-hours/week) than in the insufficient active group (<10 Met-hours/week).

In this study, we observed a significant negative association between PA and HbA1c only after taking initial levels PA and HbA1c into consideration. In stratified analysis by baseline levels of PA and HbA1c, we observed a negative association between PA and HbA1c level only among patients with a lower level of baseline PA and HbA1c. It appears that patients with more than 59.5 MET-hours/week of PA had already reached the functional threshold of PA and could not benefit more from an additional increase in PA in glycemic control.

The strengths of this study included the longitudinal design and large sample size, which enable us to evaluate the effect of PA directly, and made it possible to conduct stratified analysis by initial PA and HbA1c level. Second, the measurement of PA across a broad spectrum of daily living domains enabled us to evaluate the effect of total PA and specific type of PA, while most previous studies just focused on the effect of leisure-time activities. Moreover, this study is among the few studies evaluating the post-intervention effects of PA on HbA1c in diabetes patients, providing evidence on the long-term beneficial effect of PA. Finally, the use of multiple level mixed regression models enables us to evaluate the independent effect of PA on HbA1c during the whole period of follow-up.

Several limitations should be mentioned. First, the baseline levels of PA and HbA1c were not comparable across the groups reclassified by PA, which may have introduced confounding effects to the results. However, stratified analyses by the two factors not only released our concern on their confounding effects but also demonstrated their potential modifying effects on PA–HbA1c associations. Second, PA level was measured based on the IPAQ, and the METs were calculated based on the average level of PA regardless of age, sex, body size, and environmental conditions in which the activities were performed. Moreover, PA level was measured repeatedly using IPAQ, which is not reliable to measure the changes in PA at the individual level [46]. Misclassification bias could not be avoided. Finally, detailed information on anti-diabetes agent use and insulin shots were not available across the whole period of follow-up. Residual confounding effect of the factors, as well as other factors such as diet and self-efficacy, could not be excluded.

## 5. Conclusions

In summary, PA level can be improved by exercise-focused intervention in Chinese T2DM patients. The benefit effect of PA on HbA1c level is dependent on baseline levels of PA and HbA1c. Our results, particularly the beneficial role of PA in improving HbA1c in patients with lower levels of PA or HbA1c, suggest that the interventions should be addressed in the physically inactive patients to improve their PA level to a functional threshold.

## Figures and Tables

**Figure 1 ijerph-18-04292-f001:**
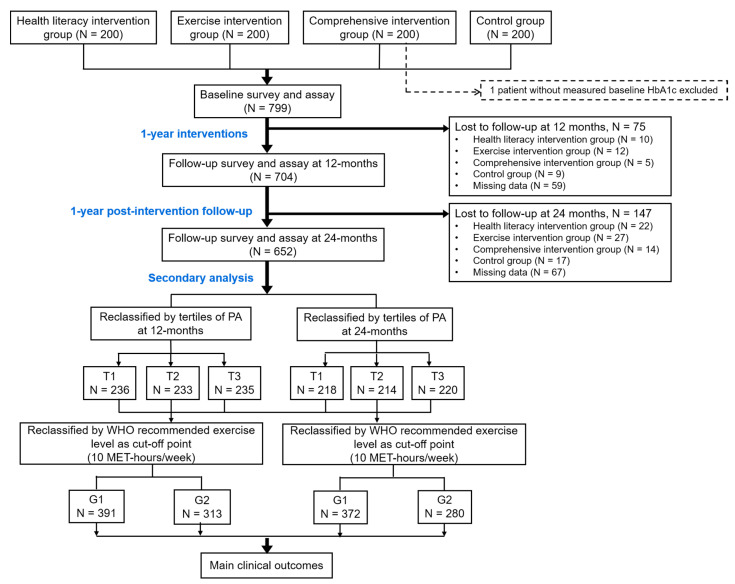
Flow diagram of the study. PA: Physical Activity; T1-T3: Group 1–3 by tertiles of PA; G1-G2: Group 1-2 by WHO (World Health Organization) recommended exercise level.

**Figure 2 ijerph-18-04292-f002:**
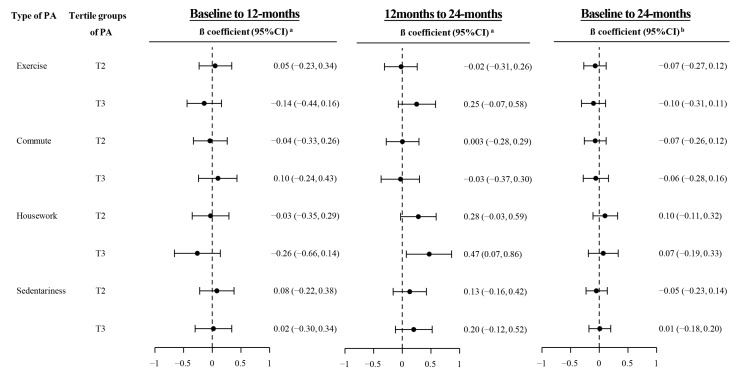
Associations of HbA1c with levels of specific type of PA (Physical Activity). ^a^ Two-level mixed regression models with individuals as the level-2 and intervention status as the level-1 observations, and adjusted for age, sex, anti-diabetes drug use or insulin use, duration of diabetes, initial levels of A1c and PA, overall PA level; ^b^ Three-level mixed regression models with individuals as the level-3, intervention status as the level-2, and repeated measurements of PA as the level-1 observations, and adjusted for age, sex, anti-diabetes drug or insulin use, duration of diabetes, baseline levels of A1c and PA, and overall PA level.

**Figure 3 ijerph-18-04292-f003:**
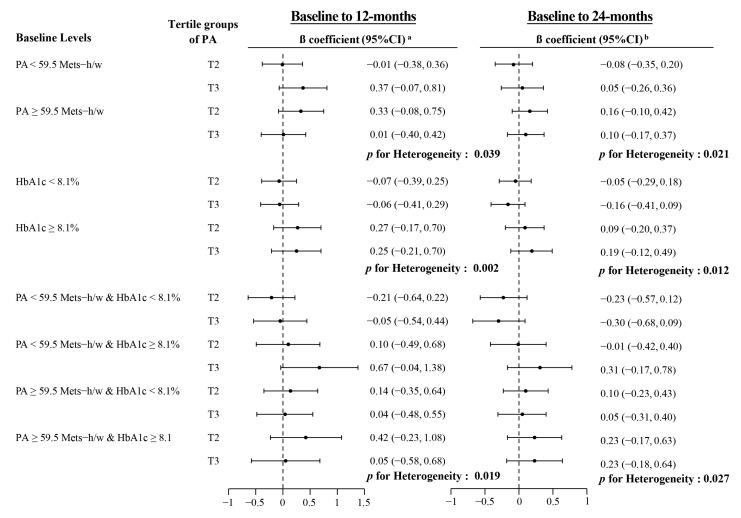
β coefficients and 95% confidence intervals for PA with HbA1c at 12 and 24 months by initial levels of PA and HbA1c. ^a^ Two-level mixed regression models with individuals as the level-2 observations and intervention status as shown in Table 1 and adjusted for age, sex, use of anti-diabetes agents and insulin, duration of diabetes, and initial PA level (for stratified analysis by initial HbA1c only) or initial HbA1c level (for stratified analysis by initial PA only); ^b^ Three-level mixed regression models with individuals as the level-3, intervention status as the level-2 and repeated measurements of PA as the level-1 observations, and adjusted for age, sex, use of anti-diabetes agents and insulin, duration of diabetes, and initial PA level (for stratified analysis by initial HbA1c only) or initial HbA1c level (for stratified analysis by initial PA only).

**Table 1 ijerph-18-04292-t001:** Physical activity (PA) levels during the study period among diabetes patients in four arms.

Median (IQR) of PA (METs-h/w)	Intervention Groups			Control Arm	*p*-Value ^a^
	Health Literacy	Exercise	Comprehensive		
At baseline					
Overall	56.0 (30.0, 84.0)	50.5 (33.5, 80.7)	50.0 (28.0, 90.0)	71.5 (53.0, 108.0)	<0.001
Exercise	0 (0, 14.0)	4.0 (0, 20.0)	0 (0, 20.0)	7.5 (0, 28.0)	0.01
Commute	18.7 (0, 30.2)	16.0 (8.0, 28.0)	12.0 (0, 28.0)	28.0 (14.0, 29.0)	<0.001
Housework	21.0 (4.5, 42.0)	21.0 (10.5, 42.0)	21.0 (10.5, 42.0)	42.0 (21.0, 52.5)	<0.001
Sedentariness	35.0 (28.0, 49.0)	31.5 (21.0, 49.0)	28.0 (14.1, 35.0)	24.5 (14.0, 31.5)	<0.001
At 12 months					
Overall	41.8 (16.9, 66.1) *	70.0 (41.5, 97.4) *	55.0 (28.0, 81.0)	73.7 (44.3, 108.3)	<0.001
Exercise	0 (0, 10.0)	12.0 (0, 22.7)	8.2 (0, 28.0)	7.3 (0, 28.0)	<0.001
Commute	10.0 (0, 28.0) *	23.3 (10.0, 38.5) *	12.0 (0, 28.0)	28.0 (6.3, 42.0)	<0.001
Housework	18.0 (5.0, 42.0) *	30.8 (9.0, 42.0)	21.0 (7.5, 42.0) *	31.5 (10.5, 42.0) *	0.001
Sedentariness	28.0 (21.0, 35.0) *	28.0 (21.6, 35.0) *	28.0 (21.0, 35.0)	28.0 (21.0, 42.0) *	0.94
At 24 months					
Overall	39.5 (17.8, 69.4) *	57.0 (29.0, 91.0) ^+^	64.8 (30.8, 104.5) ^+^	70.0 (36.0, 110.0)	<0.001
Exercise	0 (0, 4.7) *^+^	4.0 (0, 20.0) ^+^	17.0 (0, 38.0) *^,+^	8.0 (0, 24.0)	<0.001
Commute	10.0 (0, 28.0) *	18.7 (8.0, 42.0) *	13.7 (0, 30.0) ^+^	14.0 (0, 30.7)	<0.001
Housework	21.0 (6.0, 42.0) ^+^	21.0 (9.0, 42.0) ^+^	21.0 (9.0, 42.0) *	31.5 (10.5, 49.0) *	0.03
Sedentariness	35.0 (21.0, 42.0) *^+^	31.5 (21.0, 45.5) ^+^	28.0 (21.0, 42.0) *	28.0 (17.5, 35.0) *	<0.001

PA: Physical Activity; IQR: interquartile range; METs-h/w: metabolic equivalents (METs)-hours per week. ^a^ ANOVA tests (normal distributed) or Kruskal–Wallis tests (non-normal distributed) for group-comparisons; * *p* < 0.05 for differences in PA level between baseline and 12 months and between baseline and 24 months within each group using a mixed linear model; ^+^
*p* < 0.05 for difference in PA level between 12 and 24 months within each group using a mixed linear model.

**Table 2 ijerph-18-04292-t002:** β and 95%CI of hemoglobin A1c (HbA1c) during the following-up with baseline characteristics of the study participants.

Baseline Characteristics	HbA1c Level (%, Median, IQR)			β Coefficients (SE)	
	At baseline	At 12-months	At 24-months	Unadjusted	Adjusted ^a^
Age (years)					
<65	8.1 (7.5, 9.0)	8.0 (6.9, 9.1)	7.9 (7.0, 9.1)	0 (ref)	0 (ref)
≥65	8.1 (7.6, 9.2)	8.0 (7.1, 9.1)	7.9 (6.9, 9.1)	0.05 (0.10)	0.06 (0.14)
Sex (%)					
Men	8.1 (7.6, 9.1)	8.1 (7.1, 9.2)	7.9 (7.0, 9.1)	0 (ref)	0 (ref)
Women	8.1 (7.5, 9.1)	8.0 (7.1, 9.1)	7.8 (6.9, 9.1)	−0.11 (0.10)	−0.11 (0.09)
Educational level (%)					
Primary school or below	8.1 (7.5, 8.9)	8.1 (7.2, 9.3)	8.1 (7.2, 9.7)	0 (ref)	0 (ref)
Junior high school	8.2 (7.6, 9.3)	8.1 (7.1, 9.2)	7.9 (7.0, 9.1)	−0.13 (0.13)	−0.23 (0.13)
Senior high school	8.1 (7.6, 8.9)	8.0 (7.1, 8.8)	7.9 (7.0, 8.9)	−0.20 (0.14)	−0.25 (0.13)
College and above	7.9 (7.4, 8.7)	7.6 (6.7, 9.1)	7.5 (6.7, 8.6)	−0.31 (0.17)	−0.29 (0.16)
Monthly income *per capita* (USD, %)					
<308	8.1 (7.5, 8.8)	8.3 (7.5, 9.5)	8.3 (7.4, 9.7)	0 (ref)	0 (ref)
308–769	8.1 (7.6, 9.2)	7.9 (6.9, 9.0)	7.8 (7.0, 9.0)	−0.32 (0.14) *	−0.40 (0.13) **
≥769	8.0 (7.5, 9.2)	8.0 (7.2, 9.1)	7.7 (6.9, 8.8)	−0.31 (0.16) *	−0.34 (0.15) *
Tobacco use (%)					
Never	8.1 (7.6, 9.1)	8.0 (7.1, 9.1)	7.9 (7.0, 9.0)	0 (ref)	0 (ref)
Ever	8.0 (7.5, 9.2)	8.2 (7.1, 9.3)	8.2 (7.1, 9.4)	0.09 (0.14)	-0.01 (0.14)
Alcohol drinking (%)					
Never	8.1 (7.6, 9.1)	8.0 (7.0, 9.1)	7.9 (6.9, 9.1)	0 (ref)	0 (ref)
Ever	8.3 (7.5, 9.2)	8.3 (7.2, 9.3)	8.3 (7.4, 9.2)	0.08 (0.15)	-0.01 (0.15)
Years of diabetes					
<10	8.0 (7.5, 8.9)	7.7 (6.9, 9.0)	7.7 (6.8, 8.9)	0 (ref)	0 (ref)
≥10	8.2 (7.7, 9.4)	8.1 (7.4, 9.3)	8.0 (7.2, 9.2)	0.37 (0.09) **	0.31 (0.09) **
Medications (%)					
Diabetes pills only	8.1 (7.5, 9.1)	7.9 (7.0, 9.1)	7.9 (6.9, 9.1)	0.09 (0.18)	0.02 (0.17)
Insulin shot only	8.1 (7.6, 9.5)	8.4 (7.3, 9.0)	8.0 (7.3, 9.2)	0.10 (0.12)	0.06 (0.11)
Neither	8.0 (7.5, 8.9)	7.7 (6.8, 9.5)	7.3 (6.7, 8.2)	0 (ref)	0 (ref)
Both	8.3 (7.8, 9.2)	8.3 (7.4, 9.4)	8.0 (7.1, 9.4)	−0.19 (0.21)	−0.10 (0.20)
c-HeLMS score					
<116	8.1 (7.6, 9.2)	8.0 (7.1, 9.1)	7.9 (6.8, 9.1)	0 (ref)	0 (ref)
≥116	8.1 (7.6, 9.1)	7.9 (7.0, 9.1)	7.8 (7.1, 9.0)	0.04 (0.10)	0.04 (0.09)
Correct rate of c-DNT-5					
<80	8.3 (7.7, 9.4)	8.0 (7.1, 9.1)	7.7 (6.8, 9.2)	0 (ref)	0 (ref)
≥80	8.1 (7.5, 9.0)	8.0 (7.1, 9.1)	7.9 (7.0, 9.0)	−0.01 (0.11)	0.11 (0.10)
PA level (Mets, by tertile)					
<42	8.2 (7.7, 9.0)	8.0 (7.1, 9.0)	8.0 (6.9, 9.1)	0 (ref)	0 (ref)
42–78	8.1 (7.5, 9.2)	7.9 (7.0, 9.2)	7.8 (7.0, 9.0)	0.04 (0.12)	0.07 (0.11)
≥78	8.1 (7.5, 9.2)	8.1 (7.1, 9.3)	7.9 (7.0, 9.0)	−0.05 (0.12)	−0.02 (0.12)

SE: Standard Error; USD: USA dollar; HeLMS: Health Literacy Management Scale; c-DNT-5: the 5-item Diabetes Numeracy Test scale; β derived from three-level mixed regression models (intervention status are level-3 observation units, individuals are level-2 and repeated measurements are level-1). ^a^ adjusted for age, sex, and baseline HbA1c; * *p* value < 0.05; ** *p* value < 0.01.

**Table 3 ijerph-18-04292-t003:** Baseline characteristics of the study participants by tertile groups of PA at the follow-up surveys.

Characteristics at Baseline	PA Level at the 12-Month Survey (METs-h/w)			*p*-Values	PA Level at the 24-Month Survey (METs-h/w)			*p*-Values
	Tertile Group 1 (<38.7, *n* = 236)	Tertile Group 2 (38.7 to 77.0, *n* = 233)	Tertile Group 3 (≥77.0, *n* = 235)		Tertile Group 1 (<38.8, *n* = 218)	Tertile Group 2 (38.8 to 82.0, *n* = 214)	Tertile Group 3 (≥82.0, *n* = 220)	
Age (years, median, IQR) ^a^	67 (60, 74)	67 (60, 72)	64 (58, 69)	<0.001	68 (60, 74)	66 (59, 70)	64 (59, 70)	0.002
Sex of men (%) ^b^	53.8	46.8	37.0	<0.001	53.7	36.5	44.1	0.045
Educational level (%) ^b^				0.118				0.432
Primary school or below	25.1	26.6	15.7		27.2	18.7	20.9	
Junior high school	35.3	39.1	40.9		34.5	39.7	40.0	
Senior high school	26.4	19.3	29.4		24.9	25.2	25.5	
College and above	12.2	15.0	14.0		13.4	16.4	13.6	
Monthly income per capita (USD, %) ^b^				0.482				0.838
<308	13.5	18.7	11.5		20.0	11.7	12.9	
308–769	53.0	59.6	61.3		49.8	58.7	62.7	
>769	33.5	21.7	27.2		30.2	29.6	24.4	
Tobacco smoking (%) ^b^	18.8	13.4	14.6	0.210	15.3	14.6	14.3	0.771
Alcohol drinking (%) ^b^	14.2	11.9	8.3	0.045	16.6	10.1	7.6	0.003
Years of diabetes (median, IQR) ^a^	10.0 (5.3, 16.4)	9.8 (5.0, 14.8)	9.4 (4.8, 15.1)	0.362	11.0 (5.5, 17.0)	8.9 (4.8, 14.9)	9.7 (5.0, 15.0)	0.026
Use of anti-diabetes agents and insulin (%) ^b^				0.024				0.660
Diabetes pills only	61.9	67.2	52.1		62.2	64.4	61.5	
Insulin shot only	10.2	4.9	9.3		10.1	7.3	8.9	
Both	24.8	23.0	49.4		22.0	23.4	22.1	
Neither	3.1	4.9	9.2		5.7	4.9	7.5	
PA level (MET/h-w, median, IQR) ^a^	49.0 (26.6, 73.0)	56.0 (33.0, 84.0)	77.0 (48.7, 112.0)	<0.0001	50.5 (24.0, 80.7)	63.0 (41.0, 90.0)	69.3 (44.4, 104.0)	<0.0001
HbA1c level (%) ^b^	8.2 (7.6, 9.2)	8.1 (7.5, 9.0)	8.2 (7.5, 9.3)	0.655	8.2 (7.6, 9.2)	8.2 (7.5, 9.2)	8.1 (7.6, 8.9)	0.567
HbA1c < 7.0% (%) ^b^	22.5	22.3	21.7	0.844	26.2	24.3	25.9	0.956
Energy Intake (Kcal, median, IQR) ^a^	1448 (1116, 1793)	1522 (1197, 1826)	1380 (1130, 1691)	0.120	1467 (1134, 1821)	1433 (1132, 1720)	1410 (1141, 1711)	0.384

^a^ Continuous variables were presented as medians and interquartile range (IQR); ^b^ Categorical variables were presented as frequency and percentages. *p*-values for one-way ANOVA or Kruskal–Wallis tests (continuous variables) and chi-square tests (categorical variables).

**Table 4 ijerph-18-04292-t004:** Associations of PA level with improvements in HbA1c among the study participants.

Associations of PA level with HbA1c	Tertile Groups by PA Level (METs-h/w) ^a^			*p*-Values	Achievement of Recommended Exercise Level		*p*-Values
	Lowest	Medium	Highest		No	Yes	
At 12 months							
Number of subjects	236	233	235		391	313	
HbA1c (%, median, IQR)	8.0 (7.1, 9.0)	7.9 (7.0, 9.2)	8.1 (7.1, 9.3)	0.561	8.0 (7.1, 9.2)	7.9 (7.0, 9.0)	0.276
HbA1c < 7.0% (*n*, %)	53 (22.5)	52 (22.3)	51 (21.7)	0.844	87 (22.3)	69 (22.0)	0.948
β1 (95%CI)	0 (ref)	0.12 (−0.16, 0.40)	0.15 (−0.15, 0.45)		0 (ref)	−0.15 (−0.39, 0.09)	
β2 (95%CI)	0 (ref)	0.15 (−0.12, 0.42)	0.06 (−0.23, 0.35)		0 (ref)	−0.05 (−0.29, 0.18)	
β3 (95%CI) ^a^	0 (ref)	−3.47 (−5.33, −1.60)	−0.85 (−2.58, 0.88)		0 (ref)	−0.28 (−0.54, −0.02)	
At 24 months							
Number of subjects	218	214	220		372	280	
HbA1c (%, median, IQR)	8.0 (6.9, 9.1)	7.9 (7.0, 8.9)	7.8 (6.9, 9.0)	0.794	7.8 (6.3, 9.0)	7.9 (6.9, 9.1)	0.816
HbA1c < 7.0% (*n*, %)	57 (26.2)	52 (24.3)	57 (25.9)	0.956	94 (25.3)	72 (25.7)	0.897
β1 (95%CI)	0 (ref)	−0.01 (−0.31, 0.29)	−0.17 (−0.47, 0.14)		0 (ref)	−0.07 (−0.33, 0.18)	
β2 (95%CI)	0 (ref)	0.03 (−0.25, 0.31)	−0.10 (−0.39, 0.19)		0 (ref)	−0.01 (−0.25, 0.23)	
β3 (95%CI)	0 (ref)	−0.04 (−0.49, 0.41)	−0.50 (−1.00, −0.01)		0 (ref)	−1.65 (−3.12, −0.18)	
From baseline to 24 months							
β (95%CI)	0 (ref)	−1.29 (−3.33, 0.75)	−3.49 (−5.87, −1.11)		0 (ref)	−0.20 (−0.38, −0.02)	

^a^ Cut-off points for tertile groups of PA were 38.7 and 77.0 Mets/h-w at 12-months, and 38.8 and 82.0 Mets/h-w at 24-months, respectively; β1, β2, and β3 derived from two-level mixed regression models with individuals as the level-2 and intervention status as the level-1 observations; β1 adjusted for age and sex; β2 additionally adjusted for use of anti-diabetes agents and insulin, duration of diabetes, and baseline levels of HbA1c and PA; β3 additionally adjusted for use of anti-diabetes agents and insulin, duration of diabetes, initial levels of HbA1c and PA, and interactions of PA with initial level of PA and HbA1c. β derived from three-level mixed regression models with individuals as the level-3, intervention status as the level-2 and repeated measurements of PA as the level-1 observations, and adjusted for age, sex, income, use of anti-diabetes agents and insulin, duration of diabetes, baseline levels of A1c and PA, and interactions of PA with baseline levels of PA and HbA1c. Reclassifying the subjects into two groups according to their leisure time activities using the WHO recommended exercise level (10 MET-hours/week) as the cut-off point observed similar results. After adjusting for potential confounders including interactive variables, a significant improvement in HbA1c level was observed for patients achieving goal exercise level versus those not from baseline to 12 months (β: −0.31, 95%CI: −0.56, −0.05), from 12 to 24 months (β:−1.65, 95%CI: −3.12, −0.18) and from baseline to 24 months (β:−0.20, 95%CI: −0.38, −0.02).

## Data Availability

The datasets used and/or analyzed during the current study are available from the corresponding author on reasonable request.

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
