# Peer review of "Physical Activity and Glycemic Control Status in Chinese Patients with Type 2 Diabetes: A Secondary Analysis of a Randomized Controlled Trial"

_ijerph, 2021, doi:10.3390/ijerph18084292_

Round 1

Reviewer 1 Report

This is a good secondary analysis from a large study from China, There are many studies showing physical activity , exercise and fitness , as well as muscle strength, improve risk of diabetes ( Wany Y et al. Am J Med 2019; 132: 1225-1232 and Wany Y et al. Mayo Clin Proc 2019; 94: 643-651.) Although this idea are well known , this shows it again in a well-done large study from China. One would expect more improvement in higher than lower baseline A1C , so the authors may speculate on the reasons for this??

Author Response

Thank you very much for your consideration and comments.

We indeed observed an improvement in HbA1c among patients with lower baseline HbA1c but not those with higher baseline HbA1c. As we included diabetes patients with the most recent A1c ≥ 7.5% (58 mmol/mol or fasting glucose level ≥ 10 mmol/L), it is possible that the patients with HbA1c ≥ 8.1% were less sensitive to pharmacological and non-pharmacological interventions including physical activity and diet.

Reviewer 2 Report

  1. I wonder how the communities were recruited , and how the communities were randomly assigned ? these need a further description in session of "Method"
  2. How did the authors deal with missing data/ loss of follow up?

Author Response

Thank you very much for your consideration and useful comments. We have revised the manuscript along the lines, as elaborated point by point below (changes are marked in red in the revised manuscript):

1) I wonder how the communities were recruited, and how the communities were randomly assigned? these need a further description in session of "Method"?

Response: We now add some detailed descriptions of the recruitment and randomization in the section of Method, please see Line 105 to 130.

2) How did the authors deal with missing data/ loss of follow up?

Response: In this analysis, we excluded all missing data, since only a small amount of data was missing (95 at the end of the intervention and 147 at 1-year post-intervention) and no significant difference was observed in demographic characteristics between participants and those lost to follow-up. We now add the information to the section of Methods (line 237-240, highlighted in red).

Reviewer 3 Report

This is an interesting study on the role of physical activity on hemoglobin A1c level in Chinese patients with type 2 diabetes mellitus. The topic is relevant, even if not very novel. The article is well written, the intervention well planned, the results well-presented and discussed. The limitations of the study were punctuated by the authors and do not compromise the publication. Therefore, the article can be published in its present form in International Journal of Environmental Research and Public Health

Author Response

Thank you very much for your attention and consideration.

Reviewer 4 Report

See attached file. 

Author Response

Thank you very much for your consideration and useful comments. We have revised the manuscript along the lines, as elaborated point by point below (changes are marked in red in the revised manuscript, please see the attachment. ) :

1) Explain PA groups better, for example when they sedentary.

Response: In the RCT, the exercise group received usual diabetes care and were asked to walk 3–5 times a week, 30–40 min per time in the first 6 months, and 60–70 min per time in the following 6 months. They could walk continuously or divide the exercise into shorter time periods, but for at least 10 min at a time, please see Line 143 to 147, which is highlighted in red.

2) Explain health literacy group.

Response: In addition to usual diabetes care, the health literacy group received guidance from the healthcare providers using a Chinese adapted version of the Partnership to Improve Diabetes Education (PRIDE) toolkit which was designed to aid healthcare provider-patient communications on diabetes management. Before the intervention, the healthcare providers were trained with a Clear Health Communication Curriculum to improve their diabetes-related counseling communication skills (Line 153 to 159, highlighted in red).

3) add table 2 glycemic control groups.

Response: We now add the glycemic control status of the patients to Table 3.

4) add information about metformin.

Response: Unfortunately, we just collected the information on the use of anti-diabetes agents and insulin shots, but not the details on any specific agent.

5) any diabetes with end damage?

Response: We did not collect information on diabetes complications or end damage.

6) could you divide by sex results?

Response: In this study, we did not observe significant heterogeneity in associations between PA and HbA1c by sex. Due to the limited sample size, further analysis stratified by sex made the statistical power not enough to evaluate the associations. We now add the information to the section of Methods (Line 240 to 242, highlighted in red).

Reviewer 5 Report

Explain PA groups better, for example when they sedentary

Explain health literacy group

add table 2 glycemic control groups

add information about metformin

any diabetes with end damage?

could you divide by sex results?

Author Response

Thank you very much for your consideration and useful comments. We have revised the manuscript along the lines, as elaborated point by point below (changes are marked in red in the revised manuscript, please see the attachment. ) :

1) Study Design and Methods: Borg RPE visual scale is a psycho-physical tool. How co-authors considered the (expected) subjectivity of it? Some studies had demonstrated Borg scale may outperform scales or items used in some other similar scenarios.

Response: It is true that Borg RPE visual scale is a psycho-physical tool to measure the intensity of PA. Considering its subjective nature, we did not use the data in this analysis. The PA levels of the subjects at baseline, 12- and 24-months were assessed using the validated International Physical Activity Questionnaire (IPAQ) (https://www.who.int/ncds/sur-veillance/steps/GPAQ_CH.pdf).

2) Statistical methods: Multiple-level mixed regression models were adjusted for age, sex, duration of diabetes, use of anti-diabetes drugs, etc. What about the dietetic therapy or dietary counseling information? If it was not possible to identify baseline dietetic items in patient´s information or in the following, it would be advisable to explain and discuss it as a study limitation. Diets could not be excluded as a confounding variable, but this was minimally mentioned in line 460 (discussion) as a limitation.

Response: We agree with the reviewer that dietary habits may affect HbA1c levels in diabetes patients. We collected the information at baseline, 12- and 24-months for all patients through 3-day 24 hours diet recall, and calculated total energy intake. We now present energy intake at baseline for all subjects (Table 3). However, additionally adjusting for energy intake at each survey did not change the estimated associations between PA and HbA1c. So we did not include energy intake in the models. We now add related information to the manuscript (Line 182 to 183, highlighted in red).

3) Results: Complementary tables: Both complementary tables need to be improved. Both tables are confusing in some of the variables presented. Some words must be replaced. For example, in Table 1, characteristics at baseline: “sex of men” could be replaced with “male” if it applies (male vs female). In the same Table 1, when n (%) is presented, numbers are in an uncomprehensive way. See: HbA1c level (%) b 8.1(7.5,9.1). The same confusion is in “Health literacy and Numeracy skill variables. Moreover, numbers are presented with a comma in between or with a -, confusing their understanding. The tables (S1 and S2) would improve if they were indicated in the statistical test regarding the p-value, even if it has already been described in the method. Follow the same format as in results tables. Results in general should improve their presentation.

Response: We now revise complementary tables according to the reviewer’s suggestion (Table S1 and S2).

5) In Table S3, title refers “changes”, is it - coefficient used to the magnitude? It would be worth expanding on its interpretation.

Response: We change the title to “β and 95%CI of HbA1c during the following-up with baseline characteristics of the study participants”, as shown in current Table 2.

6) I think this Table S3 would not be a supplementary one, it is more relevant for results.

Response: We now move Table S3 to the main text as Table 2.

7) Line 244 through line 248, results of variable analysis of differences are not seen in Table 1.

Response: The results were shown in Table S1. We now correct the typo (Line 249).
